# Targeting Invasion: The Role of MMP-2 and MMP-9 Inhibition in Colorectal Cancer Therapy

**DOI:** 10.3390/biom15010035

**Published:** 2024-12-30

**Authors:** Alireza Shoari, Arghavan Ashja Ardalan, Alexandra M. Dimesa, Mathew A. Coban

**Affiliations:** 1Department of Cancer Biology, Mayo Clinic, Jacksonville, FL 32224, USA; dimesa.alexandra@mayo.edu; 2Department of Pharmacy and Biotechnology, Alma Mater Studiorum, University of Bologna, 40126 Bologna, Italy; arghavan.ashja@studio.unibo.it

**Keywords:** colorectal cancer, gelatinase inhibition, MMP-2, MMP-9, matrix metalloproteinases, cancer therapy

## Abstract

Colorectal cancer (CRC) remains one of the most prevalent and lethal cancers worldwide, prompting ongoing research into innovative therapeutic strategies. This review aims to systematically evaluate the role of gelatinases, specifically MMP-2 and MMP-9, as therapeutic targets in CRC, providing a critical analysis of their potential to improve patient outcomes. Gelatinases, specifically MMP-2 and MMP-9, play critical roles in the processes of tumor growth, invasion, and metastasis. Their expression and activity are significantly elevated in CRC, correlating with poor prognosis and lower survival rates. This review provides a comprehensive overview of the pathophysiological roles of gelatinases in CRC, highlighting their contribution to tumor microenvironment modulation, angiogenesis, and the metastatic cascade. We also critically evaluate recent advancements in the development of gelatinase inhibitors, including small molecule inhibitors, natural compounds, and novel therapeutic approaches like gene silencing techniques. Challenges such as nonspecificity, adverse side effects, and resistance mechanisms are discussed. We explore the potential of gelatinase inhibition in combination therapies, particularly with conventional chemotherapy and emerging targeted treatments, to enhance therapeutic efficacy and overcome resistance. The novelty of this review lies in its integration of recent findings on diverse inhibition strategies with insights into their clinical relevance, offering a roadmap for future research. By addressing the limitations of current approaches and proposing novel strategies, this review underscores the potential of gelatinase inhibitors in CRC prevention and therapy, inspiring further exploration in this promising area of oncological treatment.

## 1. Introduction

CRC, also known as colon cancer or bowel cancer, involves cancerous growths in the colon, rectum, and appendix. It is one of the most common types of cancer diagnosed globally [1]. In the late 1990s, CRC was the fourth-leading cause of cancer death in both men and women under 50. Today, it is the leading cause of cancer death in men and the second leading cause in women [2]. CRC can result from both genetic and lifestyle factors. Key risk factors include older age, male sex, high intake of fat, alcohol, red meat, processed meats, low fiber diet, obesity, smoking, and a sedentary lifestyle [3,4]. Genetic predispositions, such as hereditary non-polyposis CRC (HNPCC) or familial adenomatous polyposis (FAP), also significantly increase risk [5]. Most CRCs begin as small, benign clumps of cells called adenomatous polyps and over time, some of these polyps become colon cancers. Polyps may be small and produce few, if any, symptoms, which is why doctors recommend regular screening tests to help prevent CRC by identifying and removing polyps before they turn into cancer [6]. Symptoms of CRC include changes in bowel habits, such as diarrhea or constipation, a feeling that your bowel does not empty completely, blood in your stool (either bright red or very dark), frequent gas pains, cramps, or bloating, weight loss for no known reason, fatigue, and nausea or vomiting [7,8].

The prevalence of CRC varies worldwide, influenced by dietary, lifestyle, and genetic factors. CRC is the third most common cancer worldwide. According to the World Health Organization (WHO), it accounts for about 10% of all cancer cases and cancer-related deaths globally [9]. The highest rates are observed in regions with more developed healthcare systems, such as North America, Europe, and parts of Asia, likely due to lifestyle factors and better reporting systems. In the United States, CRC is the third most common cancer diagnosed in both men and women, excluding skin cancers [10,11]. The American Cancer Society estimates that annually, there are over 140,000 new cases and approximately 50,000 deaths due to CRC [12]. Europe also sees a high incidence of CRC, with Western European countries showing higher rates than Eastern European countries. This variation within Europe is attributed to differences in dietary patterns and healthcare access [13]. Asian countries have a lower incidence of CRC compared to Western countries, though rates are increasing with changes in diet and lifestyle, particularly in urban areas [14]. The prevalence is also linked to age, with the majority of cases occurring in individuals aged 50 and older. This is why many countries recommend regular screening starting at age 50, which has been effective in reducing the prevalence of CRC through the detection and removal of precancerous polyps [15].

Diagnosis typically involves a combination of physical examinations, blood tests (such as a test for anemia), fecal occult blood tests, endoscopic procedures like colonoscopy, and imaging studies such as CT scans [15]. Treatment options depend on the stage of the cancer, the location of the cancer in the colon, and the patient’s overall health and preferences. Common treatments include surgery to remove the cancer, chemotherapy, radiation therapy, and targeted therapy to address specific genetic markers that may be present [16]. Preventative measures include maintaining a healthy lifestyle, diet, regular physical activity, avoiding smoking and excessive alcohol use, and undergoing regular screenings. Early detection through screening is crucial since it significantly improves the prognosis [17].

## 2. Matrix Metalloproteinases (MMPs)

The complex interplay between CRC progression and its molecular underpinnings underscores the role of MMPs, particularly gelatinases, as critical mediators of extracellular matrix remodeling. MMPs are a family of zinc-dependent endopeptidases that play a crucial role in the degradation of extracellular matrix (ECM) components [18]. These enzymes are vital for various physiological processes, including tissue remodeling, inflammation, and wound healing [19]. MMPs are characterized by several structural domains such as the propeptide domain, which keeps the enzyme in an inactive form (pro-MMP) until it is cleaved for activation, and the catalytic domain, which is the zinc binding site that contains a highly conserved zinc-binding motif, typically characterized by the sequence ’HEXXHXXGXXH’, where ’H’ stands for histidine residues that coordinate with a zinc ion, and ’E’ is a glutamic acid which also plays a role in the catalytic mechanism [20]. The zinc ion is essential for the hydrolytic activity of MMPs on peptide bonds within ECM proteins and the hemopexin-like domain in C-terminal is involved in substrate specificity and interaction with tissue inhibitors of metalloproteinases (TIMPs), which regulate MMP activity [21].

The primary function of MMPs is to break down extracellular matrix proteins, which is crucial for normal physiological processes; MMPs are involved in developmental processes, tissue repair, and remodeling and they help in the formation of new blood vessels by degrading the ECM, providing a pathway for endothelial cells [22]. MMPs facilitate the migration of immune cells by modifying the vascular and interstitial barriers and they are implicated in a range of pathological conditions, including arthritis, cardiovascular diseases, and particularly cancer [23]. These enzymes contribute significantly to cancer progression by promoting tumor growth, invasion, and metastasis and they can degrade basement membranes and other structures of the ECM which facilitating the escape of cancer cells from primary tumors and their spread to distant organs [24]. By degrading ECM components, MMPs release vascular endothelial growth factor (VEGF) and other factors, enhancing angiogenesis and supplying tumors with necessary nutrients and oxygen. MMPs alter the tumor microenvironment, influencing inflammation, immune surveillance, and cancer cell behavior [25].

Given their role in cancer progression, MMPs have been targets for cancer therapy; however, the therapeutic targeting of MMPs has been challenging due to the redundancy of their functions and the broad spectrum of their substrates, which has led to side effects in clinical trials [26]. Developing specific MMP inhibitors or modulating their activity remains a focus of ongoing research to harness their potential in cancer treatment more effectively [26].

Gelatinases are a specific subgroup within the family of MMPs that primarily degrade gelatin and collagen, two major components of the ECM, and the main gelatinases in this family are MMP-2 (gelatinase A) and MMP-9 (gelatinase B) [27]. MMP-2 and MMP-9 share structural similarities with other MMPs, including a signal peptide, propeptide, catalytic domain with a zinc-binding site, and a hemopexin-like domain (Figure 1). Unique to gelatinases, they possess three fibronectin type II-like domains inserted in their catalytic domain, which are crucial for their ability to bind and degrade gelatin and collagen types IV, V, VII, and X [28]. Gelatinases play a critical role in the remodeling of the ECM by breaking down gelatin and collagens and this activity is essential not only during normal physiological processes like wound healing and angiogenesis but also in pathological states [29]. By altering the ECM, gelatinases affect various cell functions, including migration, proliferation, and differentiation and they facilitate tumor invasion and metastasis by degrading type IV collagen, a major component of basement membranes surrounding tissues and organs [30]. Due to their crucial role in disease processes, gelatinases have been targeted by various MMP inhibitors to control tumor metastasis, inflammation, and tissue destruction; however, clinical outcomes have so far been mixed, partly due to the broad range of essential physiological functions they perform, which complicates the use of broad-spectrum MMP inhibitors [31]. Elevated levels of MMP-2 and MMP-9 are often used as biomarkers for the diagnosis and prognosis of various diseases, including cancer and cardiovascular diseases [31]. The research into gelatinases continues to be a critical area of focus, not only to better understand their role in physiological and pathological processes but also to develop more specific and effective therapeutic strategies that can target these enzymes without significant side effects.

## 3. Gelatinase’s Role in CRC Progression

Among the MMP family, gelatinases (MMP-2 and MMP-9) are particularly noteworthy due to their specificity for gelatin and type IV collagen, key components of the basement membrane disrupted during metastasis. Gelatinases, MMP-2 and MMP-9, play significant roles in the progression of CRC by facilitating several critical processes. Their involvement is largely due to their ability to degrade components of the basement membrane and extracellular matrix, thereby impacting tumor growth, invasion, and metastasis (Figure 2) [32]. Gelatinases degrade various components of the extracellular matrix, leading to changes in the tumor microenvironment and this remodeling supports tumor growth by allowing cancer cells more space to proliferate and by modifying the interaction between cancer cells and their surrounding stroma [33]. By degrading the ECM, gelatinases can release and activate growth factors that were sequestered within the matrix and these growth factors, such as transforming growth factor-beta (TGF-β) and VEGF, promote tumor growth and angiogenesis [32]. MMP-2 and MMP-9 facilitate angiogenesis, which is critical for tumor growth beyond a certain size by providing necessary nutrients and oxygen; these enzymes do this by breaking down the ECM components around blood vessels, thus allowing endothelial cells to migrate and form new blood vessels [34]. One of the key functions of gelatinases in CRC is the degradation of type IV collagen, a major component of basement membranes which this degradation is essential for the invasion of cancer cells into surrounding tissues and for their entry into the bloodstream, leading to metastasis [32]. By modifying the physical barrier of the ECM and altering cell–matrix adhesions, gelatinases enable cancer cells to move more freely, and this increased motility is crucial for the invasive capacity of cancer cells [35]. The remodeling of the ECM by MMP-2 and MMP-9 can also modulate the local immune response, potentially aiding the tumor in evading immune surveillance and changes in the ECM can influence the infiltration and activity of immune cells in the tumor microenvironment [36,37].

Given their critical roles in CRC progression, gelatinases have been studied as potential therapeutic targets. However, the challenge has been to inhibit their activity selectively without disrupting their normal physiological functions. Additionally, levels of MMP-2 and MMP-9 are often examined in patient tissues and fluids as potential biomarkers for the diagnosis, prognosis, and monitoring of CRC [38]. The inhibition of MMP-2 and MMP-9 is considered a promising therapeutic strategy in CRC and other kinds of cancer [39] and can elicit significant consequences such as reducing tumor growth and angiogenesis, limiting invasion and metastasis, improving clinical outcomes, overcoming resistance mechanisms, and possibility for combinatorial treatment strategy [40]. Despite the theoretical and observed preclinical benefits of gelatinase inhibitors in CRC, clinical outcomes have been mixed. Early clinical trials with broad-spectrum MMP inhibitors have often failed due to lack of efficacy and significant side effects; however, newer strategies focus on developing more selective inhibitors that target gelatinases specifically, aiming to minimize side effects while maximizing anti-tumor activity [41]. Cancer treatment resistance is a major hurdle in effective therapy management. Since MMPs, including gelatinases, are involved in the development of resistance mechanisms (e.g., through modulation of the tumor microenvironment and interaction with other cellular pathways), targeting these enzymes could potentially overcome or reduce resistance to other therapies, such as chemotherapy and radiation [42]. Given the multifunctional nature of gelatinases in tumor progression, combining gelatinase inhibitors with other therapeutic modalities can be a strategic approach; for example, using gelatinase inhibitors in combination with chemotherapy or targeted agents might enhance the overall therapeutic efficacy and manage drug resistance more effectively [43]. Monitoring levels of MMP-2 and MMP-9 in patients could serve as a biomarker to assess the aggressiveness of the tumor, predict prognosis, or evaluate the effectiveness of gelatinase inhibitors as part of the treatment regimen [44]. Recognizing the pathological significance of gelatinases in CRC progression has spurred the development of targeted therapeutic approaches aimed at mitigating their activity. In the subsequent sections, we will provide examples of gelatinase inhibitors used in the treatment of CRC, categorized as natural, chemical small molecules, microRNA, and protein-based inhibitors (Figure 3). Natural inhibitors, such as flavonoids and other plant-derived compounds, have demonstrated effectiveness in suppressing gelatinase activity. Additionally, microRNAs can indirectly regulate the expression of these enzymes, offering a gene-based approach to inhibition. Protein-based inhibitors directly bind to and deactivate gelatinases, thereby impeding cancer progression. These diverse approaches highlight the multifaceted potential of targeting gelatinases in CRC treatment.

## 4. Natural-Based Gelatinase Inhibitors

Natural compounds have gained considerable attention in the search for MMP inhibitors due to their structural diversity, biocompatibility, and often reduced side effects compared to synthetic drugs. Natural-based anti-cancer agents are compounds derived from natural sources such as plants, animals, fungi, and marine organisms that exhibit potential to prevent, inhibit, or treat cancer [45]. The appeal of these agents lies in their vast chemical diversity and their often-lower side effects compared to conventional chemotherapy and many chemotherapeutic agents are derived from plant compounds [46]. For instance, paclitaxel (Taxol) is derived from the bark of the Pacific yew tree and is used to treat various cancers including breast, ovarian, and lung cancer [47]. Many of these agents interfere with microtubule function and prevent cancer cells from dividing. Some natural compounds can trigger non-apoptotic cell death in cancer cells, a pathway that is often defective in cancer [48]. These agents can prevent the angiogenesis that tumors need to grow, and some natural agents can damage or interfere with DNA replication in cancer cells, leading to cell death [49].

Natural-based gelatinase inhibitors are an area of keen interest in the treatment of CRC due to their potential to inhibit tumor growth and metastasis (Table 1). Inhibiting these enzymes can therefore help control the progression of cancer. Natural gelatinase inhibitors can be derived from a variety of sources, including certain phytochemicals, such as flavonoids, terpenoids, and tannins, and have shown properties that inhibit MMPs. Compounds isolated from marine algae, sponges, and other marine organisms have demonstrated potential as MMP inhibitors and some fungal and bacterial metabolites also exhibit gelatinase inhibitory activity [50]. The mechanism by which these natural agents inhibit gelatinases generally involves direct inhibition of the enzyme activity by binding to the MMPs in a manner that prevents them from interacting with their substrates, downregulation of MMP expression, often through modulation of signaling pathways within the cancer cells that control MMP gene transcription, and blocking the activation of pro-MMPs and signaling pathways to their active forms [51].

Almost two decades ago, Ko and colleagues evaluated 36 flavonoids for developing CRC treatment. Among all tested flavonoids, myricetin was identified as the most effective inhibitor of MMP-2 enzyme activity in COLO 205 cells, with an IC_50_ value of 7.82 µmol/L. Additionally, myricetin showed inhibitory effects on MMP-2 enzyme activity in various human CRC cell lines including COLO 320HSR, COLO 320DM, HT-29, and COLO 205-X, with IC_50_ values of 11.18, 11.56, 13.25, and 23.51 µmol/L, respectively. Moreover, myricetin blocked 12-O-tetradecanoylphorbol-13-acetate (TPA)-stimulated invasion of COLO 205 cells in an in vitro invasion assay [52]. Chrysin and daidzein are both naturally occurring flavonoids found in various plants and are known for their potential health benefits. Following the induction of CRC using 1,2-dimethylhydrazine (DMH) and 2% dextran sodium sulfate (DSS) for 16 weeks, rats were subjected to treatments with chrysin and daidzein at various dosages for eight weeks. The initial treatment with DMH and DSS led to a threefold increase in MMP-9 levels compared to normal rats. Subsequent treatment with chrysin at dosages of 125 and 150 mg/kg, and daidzein at dosages of 5 and 10 mg/kg, significantly reduced the MMP-9 levels in the colon. Furthermore, chrysin and daidzein showed anti-invasion activity versus SW620 cells [69]. Hawthorn proanthocyanidin refers to a group of antioxidant compounds found in the hawthorn plant. These compounds are a type of proanthocyanidin, which are a class of flavonoids known for their potential health benefits [75]. The findings from a very recent study indicated that when HCT 116 cells were treated with high-dose hawthorn proanthocyanidin extract (HPOE), for 48 h, there was a significant increase in E-cadherin protein levels compared to the control group. On the other hand, the levels of N-cadherin and MMP-9 were significantly reduced in the group receiving the high dose of HPOE, which led to an inhibitory effect on HCT 116 cell migration [74].

Norcantharidin is a chemical compound derived from cantharidin, a substance found in blister beetles (*Mylabris phalerata Pall.*). It has a similar chemical structure but differs in that it lacks a carbonyl group, making it a less toxic derivative. Norcantharidin has been studied for its potential therapeutic applications, particularly in cancer treatment [76]. Norcantharidin not only reduced MMP-9 mRNA and protein levels but also inhibited gelatinase activity in CT-26 cells in a concentration- and time-dependent manner. Further research revealed that norcantharidin decreases MMP-9 expression by suppressing Sp1 transcriptional activity [58].

Shimizu et al. demonstrated that when SW837 cells were exposed to 20 µg/mL of (−)-epigallocatechin-3-gallate (EGCG), the major biologically active component of green tea, a significant reduction in the cellular levels of MMP-9 mRNAs was observed at 12 h, with further declines noted between 24 and 48 h [53]. In similar study, EGCG (100 µg/mL) suppressed the proliferation and migration of SW620 cells (treated with PAR2-AP or factor VIIa) via inhibition of MMP-9 secretion [54].

Andrographolide is a compound derived from the leaves and stems of *Andrographis paniculata*, a plant that is widely used in traditional medicine across Asia. It is known for its anti-inflammatory, antiviral, and antioxidant properties [77]. In a study by Zhang et al., andrographolide was observed to inhibit cell proliferation, enhance cytotoxicity, and induce apoptosis in human colon cancer SW620 cells. The findings indicated that the anti-proliferative effects of andrographolide on SW620 cells were linked to the suppression of NF-κB-p65 and MMP-9 signaling pathways [62].

Resveratrol is a naturally occurring polyphenol found in various plants. One study demonstrated anti-tumorigenic effects of resveratrol in TNF-β- or TNF-α-stimulated CRC cells, as evident from MTT and invasion assays. It was shown that resveratrol treatment suppressed the activation of NF-κB induced by TNF-β or TNF-α, along with the expression of NF-κB-mediated proteins. Additionally, the anti-proliferative and anti-cancer properties of resveratrol in TNF-β-stimulated HCT 116 cells were associated with the inhibition of various gene products related to cell proliferation, invasion, and survival, such as MMP-9 [66].

Similarly, other polyphenolic compounds, such as curcumin, demonstrate broad-spectrum anti-cancer properties, including inhibition of MMP-2 and MMP-9. Curcumin, also known as diferuloylmethane, is the principal curcuminoid found in turmeric and it is known for its potent anti-inflammatory and antioxidant properties, making it a subject of interest for its potential health benefits and therapeutic applications [78]. Curcumin has demonstrated effectiveness in inhibiting MMP-2 and MMP-9 enzymes and has been shown to block the invasion of COLO 205 cells [55]. In similar study, curcumin suppressed the growth of human CRC cell lines, enhanced the apoptosis induced by capecitabine, and blocked MMP-9 activation and gene production by inhibiting the NF-κB (Nuclear factor-kappa B) pathway. In experiments with nude mice, using curcumin in combination with capecitabine proved more effective in reducing tumor volume than using either agent by itself [56]. In a similar way, aloe emodin, a natural anthraquinone, inhibited the nuclear translocation and DNA binding of NF-κB and downregulated mRNA and consequently decreased expression of MMP-2 and MMP-9, which led to blocking the phorbol-12-myristyl-13-acetate (PMA)-induced migration and invasion of WiDr colon adenocarcinoma cells [60]. In another study, curcumin was shown to inhibit the adhesion, proliferation, and cell invasion of LoVo and SW480 cells in a dose-dependent manner. The findings indicated that curcumin significantly reduced the expression of uPA and MMP9, as well as the DNA binding activity of NF-κB. Additionally, curcumin lowered the levels of the p65 subunit of NF-κB that binds to the promoter regions of the genes for uPA and MMP9, thereby inhibiting their transcriptional activation [79]. In another research on SW480, the combination of brassinin (naturally occurring phytoalexin) and imatinib (tyrosine kinase inhibitor) significantly enhanced cytotoxicity beyond what was observed with each compound alone, and also arrested the cell cycle in the G_0_/G_1_ phase. Through Annexin V binding and fluorescence imaging assays, it was demonstrated that this combination induces apoptosis in a dose-dependent manner. Additionally, the research marked the first assessment of brassinin’s impact on MMP-9 activity in SW480 cells, revealing a substantial reduction in MMP-9 activity. Moreover, the brassinin–imatinib combination was more effective in inhibiting both MMP-9 activity and its gene expression compared to the control and individual treatments [65].

Soybean saponin is a type of natural compound found in soybeans. Saponins are glycosides and they are known for their soap-like foaming properties when dissolved in water [80]. In 2008, Kang et al. revealed that soybean saponin treatment reduced mRNA expression and the secretion of MMP-2 and MMP-9 in HT-1080 cells, while it increased TIMP-2 secretion in a dose-dependent manner. Additionally, soybean saponin significantly curtailed the invasion of HT-1080 cells through a Matrigel-coated membrane. The anti-metastatic effects of soybean saponin were further validated through an in vivo experiment, which involved feeding mice with soybean saponin and then injecting them with CT-26 colon cancer cells via the tail vein. This led to a moderate reduction in the occurrence of metastatic lung tumors in the mice two weeks after the injection [57]. The further study showed that proteins in soy exhibited IC_50_ values that were 100 times lower than those of non-protein extracts, indicating that proteins may be more potent inhibitors of MMP-9 than non-protein substances. At the determined IC_50_ concentrations, non-protein fractions were more effective at reducing HT-29 cell migration and proliferation, though not via MMP-9 inhibition. In contrast, protein fractions were specifically effective in inhibiting MMP-9 activity. Overall, these findings suggest that protein fractions in soybeans may play a more significant role in cancer prevention related to soy as MMP-9 inhibitors than previously thought [81].

A study by Shin and collogues exhibited that the anthocyanins, a type of flavonoid, decreased MMP-2 and MMP-9 activities in time- and concentration-dependent manner, which related to a concurrent downregulation of their protein levels. These findings indicate that the anti-invasive properties of anthocyanins are linked to the suppression of MMP-2 and MMP-9 protein expression and activity in HCT 116 cells [59].

Brefeldin A (BFA) is a lactone antibiotic and antiviral drug that was initially isolated from the fungus *Eupenicillium brefeldianum*. Tseng et al.’s research examined the suppressive impact of BFA on human CRC cells, specifically Colo 205 cells. They discovered that BFA significantly decreased the viability of these cells in suspension (IC_50_ ≈ 15 ng/mL) by promoting apoptosis. It also hindered the clonogenic potential of Colo 205 cancer stem cells (CSCs) in both tumorsphere and soft agar colony formation assays, with similar nanogram per milliliter effectiveness. Additionally, they observed that BFA diminished the activity of MMP-9 [61].

Interesting research focused on evaluating the inhibitory effects of major seed protein fractions from eight different legume species on MMP-9 activity in colon carcinoma cells. Albumin and globulin fractions were tested for their ability to inhibit MMP-9, utilizing a fluorometric assay and gelatin zymography. Seed proteins were found to contain powerful MMP-9 inhibitors, especially those of low molecular weight. Their efficacy varied significantly across species, and a positive correlation was observed between their inhibitory effect and the reduction in cell migration. Among the legumes tested, lupin seeds were the most effective in inhibiting MMP-9, affecting both gelatinase activity and the migration and growth of HT-29 cells, whereas pea seeds exhibited no inhibitory effect [82].

Cryptotanshinone (CPT) is a natural compound extracted from the roots of *Salvia miltiorrhiza*, which is commonly used in traditional Chinese medicine. This compound is recognized for its various pharmacological properties, including anti-inflammatory, antioxidant, and anti-cancer effects [83]. In a study by Zhang et al., CPT effectively inhibited CT-26 cell invasion in vitro, reduced the protein levels of MMP-2 and MMP-9, and increased TIMP-1 and TIMP-2, both in vitro and in vivo. Additionally, CPT curbed tumor cell-induced angiogenesis in endothelial cells in vitro and rat aortic ring angiogenesis ex vivo, likely through the suppression of angiogenesis-related factors. CPT also diminished the expression of inflammatory factors in both settings. Mechanistic studies indicated that CPT blocked the PI3K/AKT/mTOR signaling pathway (Figure 4) [63].

Rosmarinic acid (RA) is a naturally occurring compound found in various plants including rosemary (*Rosmarinus officinalis*), sage, mint, and basil. It is known for its antioxidant, anti-inflammatory, and antimicrobial properties [84]. In interesting research by Han and collogues, RA reduced the growth of CRC cells by causing cell cycle arrest and apoptosis. It also hindered the invasion and migration of CRC cells and decreased the levels of MMP-2 and MMP-9. RA affected several metastatic characteristics of CRC cells by controlling epithelial–mesenchymal transition (EMT) via the increased expression of an epithelial marker. The influence of RA on EMT and MMPs was linked to the activation of AMP-activated protein kinase (AMPK). Additionally, RA prevented lung metastasis of the CRC cells by activating AMPK in a mouse model [64]. In another study, tannic acid, which is a complex polyphenol found naturally in several plant sources, was shown to suppress the survival, colony formation, and migration of the SW48 cell line. Additionally, tannic acid boosted the levels of pro-apoptotic proteins like Bim and significantly reduced the levels of MMP-9 expression, which was considered related to tannic acid’s role in inhibiting metastasis [71].

Silibinin, also known as silybin, is the major active constituent of silymarin, a standardized extract of the milk thistle seeds. Silibinin itself is extensively studied for its hepatoprotective, antioxidant, anti-inflammatory, and potentially anti-cancer properties [85]. A study by Zare et al. revealed that silibinin reduced the viability of HT-29 cell line cells in a dose-dependent manner after 24 h. TGF-β was found to elevate the mRNA and protein levels of MMP-2 and MMP-9 compared to control levels. The use of silibinin significantly inhibited these TGF-β-induced increases in MMP-2 and MMP-9 mRNA and protein expression [68].

Sauchinone is a pharmacologically active lignan isolated from the plant *Saururus chinensis*. It exhibits a range of biological effects including anti-inflammatory, antioxidant, anti-obesity, and anti-diabetic properties [86]. Transwell assays were used to assess the impact of Sauchinone on the invasive capabilities of SW480 and HCT 116 cells. The results indicated that Sauchinone significantly reduced the mobility of both cell types at concentrations of 25 and 50 μM. Further analysis was conducted on cells treated with Sauchinone to explore the underlying mechanisms; this involved measuring MMP-2 and MMP-9 expression through Western blotting. The expression levels of MMP-2 and MMP-9 were found to decrease in a concentration-dependent manner with Sauchinone treatments of 12.5 and 25 μM [70].

Triptolide is a bioactive compound derived from the *Tripterygium wilfordii* plant and it is recognized for its potent anti-inflammatory and immunosuppressive properties and has been extensively studied for its therapeutic potential [87]. Triptolide was found to significantly decrease the proliferation and invasion abilities of HT-29 cells, while also enhancing apoptosis, with these effects being more pronounced at higher concentrations. Furthermore, when Nrf2 expression was suppressed, the inhibition of proliferation and invasion in HT-29 cells was even more pronounced, and apoptosis was further increased. Additionally, the reduction in Nrf2 expression led to lower levels of the proteins MMP-2 and MMP-9, suggesting a mechanism by which triptolide may exert its effects [72].

Maple syrup is a natural sweetener made from the sap of maple trees, primarily sugar maples. Maple syrup contains various antioxidants and compounds that have been studied for their potential health benefits, including anti-cancer properties [88]. Treatment with the protein fraction of maple syrup (MSpf) showed a possible anti-tumor effect. MSpf-treated DLD-1 colon adenocarcinoma cells displayed notable reductions in proliferation, migration, and invasion. Moreover, after being treated with MSpf, there was an increase in E-cadherin levels and a decrease in MMP-9 expression levels. The findings indicated that advanced glycation end products in MSpf inhibited cell proliferation and epithelial–mesenchymal transition by blocking the STAT3 signaling pathway [89].

Punicalagin is a type of polyphenol antioxidant found in pomegranate (*Punica granatum*). It belongs to the class of compounds known as ellagitannins. Punicalagin is primarily known for its health-promoting properties, including anti-inflammatory, antioxidant, and anti-carcinogenic effects [90]. The cell viability analysis indicated that punicalagin exhibited cytotoxic effects on colon cancer cells (HCT 116, HT-29, and LoVo) in both a dose- and time-dependent fashion, while sparing normal cells. Moreover, punicalagin triggered apoptosis in colon cancer cells, as evidenced by the increased proportion of CRC cells undergoing both early and late stages of apoptosis. Additionally, the treatment with punicalagin led to a suppression of MMP-2 and MMP-9 expression in colon cancer cells [73].

The polysaccharide fraction from *Diospyros kaki* (PLE0), commonly known as the persimmon plant, is of interest due to its potential health benefits. Very recent research showed that PLE0 also suppressed the expression and enzymatic activity of MMP-2 and MMP-9, while simultaneously increasing the protein and mRNA levels of TIMP-1 in HT-29 and HCT 116 human colon cancer cells. Treatment with PLE0 for 24 h significantly reduced the number of invasive cells compared to the control cells. This evidence suggests that PLE0 decreases the metastatic potential of HT-29 and HCT 116 cells while maintaining cell viability [91].

Marine drugs, derived from marine organisms, are increasingly recognized for their potential in cancer treatment. These bioactive compounds exhibit diverse structures and mechanisms of action, making them promising candidates for developing new anti-cancer therapies [92]. Fucoxanthin is a natural carotenoid found in brown seaweeds and certain microalgae. Fucoxanthin significantly inhibited the proliferation of SW-620 cells, leading to halted growth and reduced invasive ability. This effect was partly due to the downregulation of MMP-9 mRNA and protein expression. Notably, fucoxanthin also strongly enhanced the anti-proliferative effect of 5-FU by modulating key characteristics of cancerous cells [67].

Plant sterols, also known as phytosterols, are naturally occurring compounds found in plants. They have a similar structure to cholesterol and are known for their ability to lower cholesterol levels. Recently, there has been growing interest in their potential role in cancer treatment and prevention [93]. β-Sitosterol (SITO) is a plant sterol with a chemical structure similar to cholesterol and it is found in various fruits, vegetables, nuts, and seeds. β-Sitosterol is known for its potential health benefits, including lowering cholesterol levels, supporting prostate health, and boosting the immune system [94]. The study by Shen et al. aimed to determine if liposomal encapsulated β-sitosterol (LS) has a greater inhibitory effect on tumor metastasis compared to SITO in a CT-26/luc lung metastasis mouse model. To evaluate therapeutic efficacy, male BALB/c mice were treated with PBS, SITO, or LS once daily for 7 days before intravenous injections of CT-26/luc cells; treatments continued twice a week post-cell inoculation for the duration of the experiment. The results showed that LS treatment provided better invasion inhibition with lower cytotoxicity than SITO at the same dose. Notably, mice treated with LS had significantly fewer metastases in the lungs and other tissues/organs compared to the control and SITO groups. Both SITO and LS reduced MMP-9 expression by 0.65- and 0.63-fold, respectively, compared to the control group [95].

Using natural-based gelatinase inhibitors for CRC treatment comes with several disadvantages and challenges and inhibiting MMP-2 and MMP-9 is a promising strategy, but there are several hurdles. Natural inhibitors may not be highly selective for gelatinases, and they might inhibit other MMPs or related enzymes, leading to off-target effects where non-selective inhibition can affect normal physiological processes, as MMPs are involved in tissue remodeling, wound healing, and other essential functions [96]. Many natural compounds have poor bioavailability, meaning they are not easily absorbed or distributed effectively in the body, and they can be rapidly metabolized and cleared from the body, reducing their therapeutic effectiveness [97]. While considered “natural”, these inhibitors can still cause adverse effects or toxicity at therapeutic doses and non-selective inhibition can lead to side effects due to disruption of normal MMP functions. Natural products can vary in potency and concentration depending on the source and extraction method. Ensuring consistent quality and potency in natural product formulations is challenging. Additionally, cancer cells may develop resistance to gelatinase inhibitors, reducing their long-term efficacy, and natural inhibitors might not be potent enough to achieve significant therapeutic effects in advanced cancers [43]. Obtaining regulatory approval for natural compounds can be complex, requiring extensive validation and clinical trials and large-scale production and commercialization of these compounds can be challenging due to variability in raw materials and extraction processes [98]. Natural inhibitors might need to be used in combination with other treatments, complicating therapy regimens and increasing the risk of drug interactions [99]. While natural-based gelatinase inhibitors hold potential for CRC treatment, their development and clinical application are fraught with challenges related to specificity, bioavailability, toxicity, variability, resistance, regulatory hurdles, and the complexity of cancer biology. Addressing these issues requires further research and development to optimize the efficacy and safety of these inhibitors.

## 5. Synthetic Small Molecules Inhibitors

Selective and non-selective small molecule gelatinase inhibitors are compounds designed to bind to and inhibit the activity of MMP-2 and MMP-9 or inhibit these enzymes in directly via suppression of upstream pathways. These inhibitors are synthesized through chemical processes, which allow for precise control over their structure and activity. The small size of these molecules enables them to penetrate tissues and reach intracellular targets efficiently and via inhibiting MMP-2 and MMP-9, these small molecules prevent the breakdown of the ECM, thereby limiting cancer cell invasion and metastasis [100].

These inhibitors can be designed to specifically target MMP-2 and MMP-9 with high affinity, reducing the likelihood of off-target effects. Many small molecule inhibitors can be administered orally, improving patient compliance and convenience compared to intravenous treatments [96]. Gelatinase inhibitors can be used alongside other therapeutic modalities, such as chemotherapy, radiation, and immunotherapy, to enhance overall treatment efficacy [101]. Targeting gelatinases can provide a novel mechanism to counteract resistance to conventional therapies, offering new hope for patients with refractory CRC (Table 2).

Colon cancer can metastasize to the lungs, and this occurs when cancer cells break away from the primary tumor in the colon and spread to the lungs through the bloodstream or lymphatic system [111]. About two decades ago, Ogata and colleagues explored the anti-tumor effects of synthetic MMPs inhibitor MMI270 versus postoperative lung metastasis from colon cancer in nude rats. The KM12SM human colon cancer cells were injected into the cecal wall and five weeks later, the cecum, including the tumor, was removed. Subsequently, 30 mg/kg of MMI270 was administered orally twice daily for either 2 or 4 weeks, starting immediately after tumor removal or beginning 2 weeks post-removal. At 7 weeks post-removal, early administration of MMI270 immediately after tumor removal significantly inhibited lung metastasis, whereas delayed administration did not. Survival rates were significantly higher in rats treated with early MMI270 administration compared to untreated rats. Additionally, no lung metastasis was detected in some rats that survived for 24 weeks with early MMI270 treatment. The relative MMP-9 activities were suppressed by the extract of the lung metastases for up to 4 h after oral administration of 30 mg/kg of MMI270. Similarly, the extract of lung metastases reduced the relative MMP-2 activity [102].

GL-V9, a flavonoid derivative chemically known as 5-hydroxy-8-methoxy-2-phenyl-7-(4-(pyrrolidin-1-yl) butoxy) 4H-chromen-4-one, is synthesized from wogonin. It has been noted for its pro-apoptotic, anti-inflammatory, anti-invasive, and anti-metastatic properties, which have shown effectiveness in breast cancer, gastric cancer, and hepatocellular carcinoma [112]. Gu et al.’s research demonstrated that GL-V9 reduces CRC cell viability, migration, and invasion in a concentration-dependent fashion. Moreover, the treatment with GL-V9 significantly decreased both the protein expression levels and activities of MMP-2 and MMP-9. Further investigation into the mechanisms involved showed that GL-V9 obstructs the PI3K/Akt signaling pathway, which is upstream of MMP-2 and MMP-9 [109].

Gefitinib is a targeted therapy drug used to treat certain types of cancer, and it is primarily known for its effectiveness in treating non-small cell lung cancer. Gefitinib inhibited the secretion and mRNA expression of MMP-9 and MMP-2 in HT-29 cells. It also reduced the cells’ ability to adhere to laminin and type IV collagen. These effects were observed at doses so low that gefitinib did not have an anti-proliferative effect or induce apoptosis [103].

Etodolac is a nonsteroidal anti-inflammatory drug used to treat pain and inflammation and it works by inhibiting cyclooxygenase (COX) enzymes, specifically COX-2 [113]. Ishizaki et al. injected Colon 26, a CRC cell line, into the spleens of CDF1 mice. Starting the following day, two types of COX-2 inhibitors (etodolac and nimesulide) were administered orally. The number of metastatic nodules on the liver surface was significantly lower in the etodolac-treated group compared to the controls, but there was no significant difference in the nimesulide-treated group. Additionally, the expression of MMP-9 mRNA was significantly lower in the etodolac group than in the controls, but not in the nimesulide group [104].

AMD3100, also known as Plerixafor, is a medication primarily used to mobilize hematopoietic stem cells for collection and subsequent autologous transplantation in patients with non-Hodgkin’s lymphoma and multiple myeloma. It was shown that SW480 cells’ viability was substantially blocked by AMD3100 in a dose-dependent manner. AMD3100 (100 and 1000 ng/mL) drastically hindered the invasion ability of SW480 cells. Treatment with AMD3100 significantly decreased the expression of MMP-9 but not MMP-2 in SW480 cells [105].

Pyrrole-imidazole polyamides are synthetic molecules designed to bind specifically to the minor groove of DNA. These compounds mimic the way natural transcription factors interact with DNA, allowing them to modulate gene expression by interfering with the binding of proteins to DNA [114]. A pyrrole-imidazole (PI) polyamide specifically designed to bind to the activator protein-1 (AP-1) site on the MMP-9 (Figure 5) promoter was developed and synthesized as a gene-silencing agent aimed at combating tumor metastases. This synthesized PI polyamide demonstrated a high selectivity for DNA binding and the MMP-9 PI polyamide was found to significantly reduce MMP-9 mRNA expression, protein levels, and enzymatic activity. Given that the liver is a common site for systemic metastases from CRC, the effects of MMP-9 PI polyamide treatment on tumor metastasis were analyzed using a well-established liver metastasis model, where CRC cells were injected into the spleens of athymic mice. Prior to in vivo analysis, HT-29 cells showed a reduction in MMP-9 enzymatic activity and cell invasiveness in vitro following MMP-9 PI polyamide treatment. Before tumor cell inoculation, 1 h treatment significantly decreased metastases compared to control groups, and PI treatment via tail vein injection starting 1 h after HT-29 cell implantation significantly decreased liver tumor metastasis compared to the control group [115].

17β-Estradiol (also known as E2) is a major estrogen sex hormone and the most potent form of estrogen in the body, and it plays a crucial role in the development and regulation of the female reproductive system and secondary sexual characteristics [116]. Epidemiological research shows that women have lower rates of CRC incidence and mortality compared to men [117]. Nevertheless, it remains unclear whether treatment with 17β-estradiol can effectively prevent cell movement in human colon cancer cells. Regarding this, Hsu et al. showed that 17β-estradiol treatment decreased MMP-2 and MMP-9 expression and cell mobility of human LoVo cancer cells by suppressing the activation of JNK1/2 and p38α MAPK signaling pathways [106,118].

Volatile anesthetics are a class of general anesthetics that are administered through inhalation. These anesthetics are in a gaseous or vaporized state at room temperature and include agents such as isoflurane, sevoflurane, and desflurane. The use of volatile anesthetics in cancer treatment is an emerging area of research. Volatile anesthetics, such as isoflurane, sevoflurane, and desflurane, have traditionally been used to induce and maintain general anesthesia during surgery. However, recent studies suggest they may also have potential effects on cancer biology [119,120]. An interesting study revealed that volatile anesthetics (sevoflurane and desflurane) reduce the release of MMP-9 from neutrophils and disrupt pathways downstream of CXCR2, but upstream of protein kinase C. Through downregulating MMP-9, these anesthetics decrease Matrigel degradation and subsequently hinder the invasion of CRC cells in vitro [107]. Another study also exhibited that sevoflurane inhibited cell migration and invasion in SW620 and HCT 116 cells in a concentration-dependent manner. Additionally, varying concentrations of sevoflurane suppressed ERK phosphorylation. The MMP-9 protein levels in SW620 and HCT 116 cells were also progressively reduced by different concentrations of sevoflurane in a concentration-dependent manner [121].

Nonsteroidal anti-inflammatory drugs (NSAIDs) are commonly used to manage pain and inflammation but, recently, there has been increasing interest in their potential role in cancer treatment; epidemiological studies have shown that regular use of NSAIDs, particularly aspirin, may reduce the risk of developing certain types of cancer, such as colorectal, breast, and prostate cancers [122]. Gungor et al. investigated the chemopreventive effects of NSAIDs on tumor incidence and angiogenesis in experimental CRC rats. 1,2-Dimethylhydrazine dihydrochloride (DMH) was used to induce cancer, and two NSAIDs, celecoxib and diclofenac, were administered orally as chemopreventive agents. Plasma levels of MMP-2 and MMP-9 were significantly elevated in the DMH-induced cancer group compared to the control group. However, compared to the DMH group, there was a statistically significant decrease in gelatinases levels in both the diclofenac and celecoxib groups. Additionally, TIMP-2 levels were significantly higher in the celecoxib-treated group compared to the control and DMH groups [108].

7-Allylamino-17-demethoxy geldanamycin (17-AAG) is a derivative of geldanamycin, an antibiotic that acts as a potent inhibitor of heat shock protein 90 (Hsp90). By binding to Hsp90, 17-AAG disrupts its function, leading to the degradation of client proteins that are crucial for the growth and survival of cancer cells [123]. The anti-proliferative, anti-metastatic, and anti-angiogenic effects of 17-AAG were examined both alone and in combination with Capecitabine (Cap) and/or Irinotecan (IR) on HT-29 cells by Zeynali-Moghaddam and colleagues. Among the double combination groups, only Cap/17-AAG demonstrated significant differences in MMP-9 gene expression and in the wound healing assay. Furthermore, a significant reduction in wound area was observed in the triple combination group, indicating an antagonistic effect. The IR/17-AAG double combination group showed downregulation of MMP-9 mRNA expression [124].

Quinoxaline is a heterocyclic aromatic organic compound consisting of a benzene ring fused to a pyrazine ring and this structure provides a versatile scaffold for the development of various pharmaceutical agents due to its ability to interact with multiple biological targets [125]. A novel series of quinoxaline-based dual MMP-9/monoamine oxidase-A (MAO-A) inhibitors was synthesized to inhibit CRC progression. All derivatives were initially screened using the MTT assay to evaluate cytotoxic effects on normal colonocytes for safety assessment, followed by evaluation of their anti-cancer potential on HCT 116 cells overexpressing MMP-9 and MAO-A. The most promising derivatives, 8, 16, 17, 19, and 28 (Figure 6), showed single digit nanomolar IC_50_ values against HCT 116 cells within their safe doses (EC100) on normal colonocytes. They reduced HCT1 16 cell migration by 73.32%, 61.29%, 21.27%, 28.82%, and 27.48%, respectively, as determined by the wound healing assay [126].

SB202190 is a selective inhibitor of p38 MAP kinase, a protein involved in cellular responses to stress and inflammation, and via inhibiting p38 MAP kinase, SB202190 can modulate various cellular processes, including apoptosis, differentiation, and the production of inflammatory cytokines [127]. Recently, in 2023, Kassassir et al. explored the effect of SB202190 on the invasive potential of CRC cells after platelet-derived microparticles (PMPs) uptake. The increased migration of HT-29, SW480, and SW620 cells through gelatin-coated membranes after PMP uptake was inhibited by SB202190. PMP incorporated elevated MMP-2 and MMP-9 protein levels in all three CRC cell lines, and SB202190 reduced the PMP-stimulated levels of MMP-2 and MMP-9 and related invasiveness in these cells. Furthermore, the increased amount of MMP-9 in the conditioned medium of HT-29, SW480, and SW620 cells after incubation with PMPs was reduced when SB202190 was added along with PMPs [110].

The use of chemical and small molecule gelatinase inhibitors in CRC therapy presents several disadvantages and challenges. These can be broadly categorized into issues related to efficacy, specificity, toxicity, resistance, delivery, and regulatory hurdles. Small molecule inhibitors may have limited effectiveness in reducing tumor growth or metastasis due to the complexity of cancer biology and the presence of multiple pathways driving tumor progression [128]. Many small molecule inhibitors have a short half-life in the body, requiring frequent dosing, which can be impractical for long-term treatment [129]. Small molecules often lack the specificity needed to exclusively target gelatinases. This can result in off-target effects that can lead to unintended interactions with other proteins, potentially causing adverse effects [100]. Many small molecule inhibitors suffer from poor bioavailability, meaning they do not adequately reach the tumor site in effective concentrations, which can be due to poor absorption, rapid metabolism, or an inability to penetrate the tumor microenvironment [130]. Effective delivery systems are needed to target the inhibitors specifically to tumor sites while sparing healthy tissues and developing such delivery systems can be complex and costly. The development of new small molecule inhibitors involves significant research and development costs, including synthesis, optimization, and testing in preclinical models, followed by multiple phases of clinical trials [131].

Despite these challenges, ongoing research is focused on improving the design, specificity, and delivery of chemical and small molecule gelatinase inhibitors. Combining these inhibitors with other therapeutic strategies, such as immunotherapy or targeted therapy, may enhance their effectiveness and reduce resistance.

## 6. MicroRNAs

MicroRNAs (miRNAs) are small non-coding RNAs, typically about 21-25 nucleotides long, that play crucial roles in regulating gene expression. They function by binding to complementary sequences on target messenger RNAs (mRNAs), usually resulting in gene silencing either through mRNA degradation or translational repression [132]. In the context of cancer therapy, miRNAs are of significant interest due to their involvement in various cellular processes such as proliferation, differentiation, apoptosis, and stress response and their dysregulation has been linked to the development and progression of many types of cancer [133]. Effective delivery of miRNA-based therapeutics to target tissues remains a challenge and various delivery systems are being explored, such as lipid-based nanoparticles, polymeric nanoparticles, and other nano-carriers can protect miRNAs from degradation and enhance their delivery to cancer cells [134]. Ensuring the stability of miRNA therapeutics in the bloodstream and their specific uptake by target cells is critical. Numerous clinical trials are underway to evaluate the safety and efficacy of miRNA-based therapies. These trials focus on various cancers, including lung cancer, liver cancer, and glioblastoma [135]. Future research aims to improve delivery systems, minimize side effects, and identify new therapeutic miRNA targets. Targeting gelatinases indirectly through miRNAs offers a promising therapeutic strategy (Table 3). In the upcoming paragraphs are some miRNAs examples known to directly or indirectly inhibit MMP-2 and MMP-9 and their potential roles in CRC treatment.

miR-34a, known to act as a tumor suppressor, targets multiple genes involved in cell proliferation, survival, and metastasis. Some of its well-known targets include genes like BCL2 (B-cell lymphoma 2), MET (hepatocyte growth factor receptor), and MYC (proto-oncogene) and it is regulated by the tumor suppressor protein p53, which is often mutated in various cancers [150]. Wu et al. recognized Fra-1 as a novel target of miR-34a and showed that miR-34a suppresses Fra-1 expression at both the protein and messenger RNA levels. Overexpression of miR-34a significantly suppressed colon cancer cell migration and invasion. This effect could be partially reversed by the forced expression of the Fra-1 transcript lacking the 3′-untranslated region. Additionally, the levels of MMP-9 were reduced in cells transfected with miR-34a [136]. Studies have shown that miR-497 is downregulated in several cancers, including colorectal, breast, lung, and gastric cancers. Restoring miR-497 levels in these cancer cells can suppress tumor growth and metastasis, highlighting its potential as a therapeutic target. In HCT 116 and SW480 cells transfected with miR-497 mimics, the levels of the tumor invasion markers MMP-2 and MMP-9 proteins were significantly reduced compared to control cells via targeting Fra-1, demonstrating that miR-497 inhibits invasion in these cells. These findings suggest that miR-497 inhibits EMT, migration, and invasion in HCT 116 and SW480 cells [144].

Combining natural compounds with miRNAs offers a promising strategy to inhibit tumor growth. This approach leverages the tumor-suppressive properties of miRNAs along with the anti-cancer effects of natural compounds [151]. Researchers observed increased miR-29b in colon cancer cells following exposure to hexane extract of American ginseng (HAG). Since miR-29b plays a role in regulating the migration of cancer cells, results revealed that HAG induces miR-29b expression to target MMP-2, thereby suppressing the migration of HCT 116, LOVO, and DLD-1 colon cancer cells [152]. In a similar study using a combination approach, CRC cells were treated with a gradient of emodin and either an overexpression plasmid for long noncoding RNA HCP5 or small interfering RNA targeting HCP5 (siHCP5). In HT-29 and HCT 116 cells, emodin suppressed viability, migration, invasion, and HCP5 expression in a dose-dependent manner. Silencing HCP5 inhibited viability, migration, and invasion, downregulated MMP-2 and MMP-9 levels [153].

A study by Li et al. showed that transfection with an miR-22 expression vector significantly reduced the viability of HCT 116 human colon cancer cells and suppressed their migration and invasion capabilities. The miR-22 inhibited the expression of T-cell lymphoma invasion and metastasis 1 (TIAM1) mRNA and protein. Additionally, miR-22 expression led to a decrease in the levels of the pro-invasive MMP-2 and MMP-9 expression [137]. Another study showed that administering an miR-22 mimic greatly reduced the proliferation, migration, and invasion of HCT 116 cells, while using an miR-22 inhibitor significantly increased their proliferation and invasion. After transfection with the miR-22 mimic, levels of NLRP3 (NLR family, pyrin domain-containing protein 3), MMP-2, and MMP-9 were notably decreased. In tumor tissues, overexpression of miR-22 also diminished the expression of NLRP3, MMP-2, and MMP-9 compared with the model group [138].

miR-195 is often dysregulated in various types of cancer, and in some cancers, it acts as a tumor suppressor by inhibiting cell proliferation and inducing apoptosis. Its expression levels can be correlated with cancer prognosis, making it a potential biomarker [154]. Wang and colleagues showed that MMP-9 expression significantly decreased in SW480 and HT-29 cells overexpressing miR-195, while it significantly increased in cells with downregulated miR-195 expression. Their outcomes propose that miR-195 may influence CRC cell invasion by modulating MMP-9 via targeting CARMA3 (Caspase Recruitment Domain Membrane-Associated Guanylate Kinase Protein 3) [139].

miR-149 is known to function as a tumor suppressor in various cancers and it can inhibit cancer cell proliferation, migration, and invasion. One study implied that miR-149 significantly inhibited the growth, migration, and invasion of CRC cells by targeting a transcription factor named Forkhead Box M1 (FOXM1). It was observed that the mRNA expression levels of MMP-2 and MMP-9 were downregulated in cells transfected with miR-149, which led to less metastatic CRC cells [140].

miR-302a plays a crucial role in the regulation of gene expression. It is a member of the miR-302/367 cluster, which is known for its involvement in various cellular processes, particularly in stem cell biology and early embryonic development [155]. An interesting study demonstrated that the mRNA level of miR-302a was significantly lower in CRC cell lines compared to normal colon epithelial cells. Increasing the expression of miR-302a inhibited the proliferation and invasion of CRC cells, and reduced the phosphorylation of Erk1/2 and Akt. Furthermore, the expression and secretion of MMP-9 and MMP-2 were notably reduced by the upregulation of miR-302a [141].

miR-206 is primarily known for its role in skeletal muscle development and regeneration. It is highly expressed in muscle tissues and is involved in muscle differentiation and myogenesis [156]. Wang et al. showed that the upregulation of miR-206 prevented cancer cell proliferation and migration, arrested the cell cycle, and triggered apoptosis in SW480 and SW620. The tumor-suppressive effect of miR-206 was observed in CRC cells, though these cells exhibited different metastatic potentials. They revealed that this anti-cancer effect may be attributed to the direct inhibition of NOTCH3 signaling and indirect interactions with MMP-9 expression [142].

miR-7 has been shown to function as a tumor suppressor in various cancers, including glioblastoma, breast cancer, and lung cancer. It can inhibit cancer cell proliferation, invasion, and metastasis by targeting several oncogenes and signaling pathways [157]. It has been shown that the overexpression of miR-7 suppressed the growth and movement of colon cancer cells. Moreover, it was found that miR-7 influences the proliferation and migration of colon cancer cells by modulating the protein levels of focal adhesion kinase (FAK). This regulation impacts the expression of MMP-2 and MMP-9 [145].

In some cancers, miR-9 promotes metastasis and invasion by targeting genes involved in cell adhesion and migration. In other cancers, it may inhibit tumor growth by targeting oncogenes. Overexpressing miR-9 suppressed transmembrane-4-L6 family 1 (TM4SF1) mRNA and protein levels, along with impairing wound healing, transwell migration, and invasion in SW480 cells. Moreover, miR-9 overexpression led to a decrease in MMP-2 and MMP-9 levels. These effects were reversed when SW480 cells were co-transfected with both miR-9 and TM4SF1 [143].

In a study by Zhang et al., overexpression of miR-875-5p in CRC cell lines markedly reduced cell proliferation, as demonstrated by cell viability assays, colony formation assays, and BrdU staining. Additionally, miR-875-5p induced apoptosis, evidenced by the increased levels of cleaved caspase-3 and decreased levels of the anti-apoptotic protein Bcl2. Furthermore, miR-875-5p impaired cellular migration and invasion by inhibiting MMP-9 [146]. In another study, Ke et al. identified miR-202-5p as a tumor-suppressive microRNA in CRC, where their findings revealed that miR-202-5p levels were significantly reduced in CRC tissues, and its lower expression correlated with poorer postoperative survival. Enhancing miR-202-5p expression suppressed CRC cell growth and metastasis. SMARCC1, a direct target of miR-202-5p, was found to promote CRC cell growth and metastasis. Further investigation indicated that miR-202-5p can inhibit MMP-9 [147]. Abnormal expression of miR-124-3p has been linked to several neurological disorders, including Alzheimer’s disease, Parkinson’s disease, and epilepsy. It is also associated with various cancers, where it can function as a tumor suppressor or oncogene depending on the context [158]. Functional assessment revealed that miR-124-3p is markedly downregulated in CRC tissues compared to adjacent normal samples. This downregulation negatively correlates with PD-L1 expression and transfecting HT-29 and SW480 cells with miR-124-3p mimics significantly decreased PD-L1 mRNA and protein levels. The proliferation of CRC cells was reduced, and their cell cycle was halted at the G1 phase due to the decrease in c-Myc expression. Additionally, apoptosis was triggered in CRC cells through the activation of both intrinsic and extrinsic mechanisms. Furthermore, miR-124 reduced MMP-9 expression, leading to the inhibition of cell motility and invasion (Figure 7) [149].

The process of miR-128 on the regulation of Ribophorin-II (RPN2) in CRC cells was investigated in the study by Zhou et al., and it was found that reduced levels of miR-128 in CRC tissues were inversely related to RPN2 expression. Higher RPN2 levels were significantly linked to poorly differentiated tumors, advanced stages, and lymph node metastasis in CRC patients. In HT-29 cells, overexpression of miR-128 decreased RPN2 mRNA and protein levels, significantly suppressing cell proliferation, migration, and invasion. Additionally, miR-128 mimic transfection led to lower levels of MMP-2 and MMP-9, while increasing levels of TIMP-2 [148].

Circular RNA (circRNA) is a type of non-coding RNA that forms a covalently closed continuous loop, distinguishing it from the more common linear RNA molecules. Unlike linear RNAs, circRNAs have no 5′ or 3′ ends. Due to their stability and regulatory functions, circRNAs are being explored as therapeutic targets and tools [159]. CircFNDC3B has been implicated in various cancers and it is involved in tumor suppression by sponging specific miRNAs that regulate oncogenes or tumor suppressor genes. A study by Zeng and colleagues showed that CircFNDC3B and circFNDC3B-enriched exosomes increased TIMP-3 expression and suppressed MMP-2 and MMP-9, which led to inhibiting tumorigenic, metastatic, and angiogenic characteristics of CRC. These effects were reversed by miR-937-5p overexpression or TIMP3 knockdown. In vivo studies showed that overexpression of circFNDC3B, circFNDC3B-enriched exosomes, or miR-937-5p knockdown inhibited CRC tumor growth, angiogenesis, and liver metastasis [160].

Effective delivery of miRNA mimics or inhibitors to target cells in the gastrointestinal tract is challenging and ensuring that miRNAs reach the intended cells without degradation by enzymes in the digestive system or clearance from the bloodstream is a major hurdle [161]. Achieving targeted delivery to cancer cells while avoiding healthy cells is critical to minimize off-target effects and reduce potential side effects. miRNAs are susceptible to degradation by RNases present in the bloodstream and other body fluids and this limits their stability and therapeutic efficacy; thus, chemical modifications and the use of delivery vehicles like nanoparticles can improve stability, but these methods also bring additional complexity and potential toxicity issues [162].

miRNAs can target multiple mRNAs, leading to unintended gene regulation; this lack of specificity can result in undesirable side effects and complicate the therapeutic application, and predicting and managing off-target effects is a significant challenge in miRNA-based therapies [163]. Exogenous miRNAs can trigger immune responses, potentially leading to inflammation or other immune-related issues; therefore, designing miRNA therapies that avoid immune activation without compromising their therapeutic function is an ongoing challenge [164].

Colorectal tumors are heterogeneous, with varying miRNA expression profiles among different patients and even within different regions of the same tumor. This heterogeneity complicates the development of universal miRNA-based treatments, and personalized approaches based on individual tumor profiles are necessary but add complexity to treatment strategies [165]. The path to regulatory approval for miRNA-based therapies is complex, involving rigorous testing for safety and efficacy, and long-term studies are needed to fully understand the potential risks and benefits, which can delay the availability of these treatments [166]. Producing miRNAs or miRNA inhibitors in sufficient quantities and ensuring their quality can be expensive and technically demanding, and the cost of treatment may be high, limiting accessibility for patients [167].

miRNA therapies may need to be combined with other treatment modalities (e.g., chemotherapy, targeted therapies) for optimal efficacy. Understanding and optimizing these combinations to avoid adverse interactions and enhance therapeutic outcomes is a complex task [168]. While miRNAs hold promise for improving the treatment of CRC, addressing these disadvantages and challenges is crucial for their successful clinical application.

## 7. Protein and Peptide Based Inhibitors

The development of protein and peptide-based anti-cancer medications represents a cutting-edge approach in oncology, aiming to offer more targeted and effective treatments with fewer side effects compared to traditional chemotherapies [169,170]. Proteins and peptides, due to their high specificity and ability to modulate various biological pathways, are increasingly being explored for their therapeutic potential against cancer [171].

Proteins and peptides can enhance the immune system’s ability to recognize and destroy cancer cells and some peptides have inherent cytotoxic effects on cancer cells [172]. These peptides can disrupt cellular membranes or interfere with critical intracellular pathways, leading to cancer cell death, and some protein- and peptide-based inhibitors can block angiogenic factors, starving the tumor of nutrients and oxygen [173]. Many protein- and peptide-based drugs function by inhibiting specific enzymes involved in cancer progression. For example, monoclonal antibodies and engineered peptides can target and inhibit kinases and other enzymes that drive cell proliferation and survival [170].

As naturally occurring inhibitors of MMPs, TIMPs bind to MMPs in a 1:1 ratio, inhibiting their activity; TIMP-2 is particularly effective against MMP-2, while TIMP-1 can inhibit MMP-9 [23]. Monoclonal antibodies and engineered fusion proteins can be designed to specifically target and inhibit MMP-2 and MMP-9, and protein and peptide based inhibitors offer a targeted approach to inhibiting gelatinases, potentially improving treatment outcomes for CRC patients [174]. Short sequences of amino acids designed to mimic the natural substrates or inhibitors of MMPs, which these peptides can be optimized for, increased binding affinity and specificity to the active sites of gelatinases [175]. Protein and peptides modified to improve their stability and bioavailability and peptidomimetics can retain the inhibitory function of natural peptides while overcoming some of the limitations related to degradation and rapid clearance [176,177]. Compared to small molecule inhibitors, protein- and peptide-based inhibitors often have fewer side effects due to their targeted action [178].

High specificity for MMP-2 and MMP-9 reduces off-target effects and toxicity, providing a more targeted approach to cancer treatment, and these inhibitors also can be used in combination with other therapies, such as chemotherapy, radiation, and immunotherapy, to enhance treatment efficacy. Protein and peptide based gelatinase inhibitors represent a promising approach for the treatment of CRC. By targeting the enzymes responsible for ECM degradation, these inhibitors can potentially reduce tumor invasion and metastasis, improving patient outcomes. Ongoing research aims to optimize these inhibitors for clinical application, addressing challenges related to stability, delivery, and production. The following paragraphs present samples of protein- and peptide-based inhibitors targeting gelatinase for the treatment of CRC.

Guanylyl cyclase C (GC-C) is an enzyme and a receptor that plays a crucial role in the regulation of intestinal fluid homeostasis and electrolyte balance. Dysregulation of GC-C activity is associated with various gastrointestinal disorders. Research suggests that GC-C may act as a tumor suppressor [179]. Activation of GC-C and the resulting increase in cGMP levels can inhibit the proliferation of CRC cells and induce apoptosis. GC-C signaling inhibited colon cancer cells’ capacity to breakdown matrix components, organize the actin cytoskeleton, spread, and seed distant organs. GC-C inhibited the formation of metastatic CRC cells in the mouse peritoneum by inhibiting their MMP-9 activity. Exogenous delivery of GC-C agonists may activate dormant GC-C signaling and target MMP-9 functions in CRC cells, as endogenous GC-C hormones are lacking in these tumors [180].

Ulinastatin is a glycoprotein derived from human urine and composed of polypeptide chains. As a protease inhibitor, it works by inhibiting the activity of various proteases such as trypsin, chymotrypsin, and other enzymes involved in inflammation. This inhibition helps reduce inflammation and tissue damage in various medical conditions [181]. Xu et al. observed a preventive effect of ulinastatin on the recurrence of colon cancer liver metastases in mice following hepatectomy. Ulinastatin was found to considerably hinder the in vitro invasive capacity of HCT 116 cells, as demonstrated by Transwell cell invasion experiments. In addition, the application of ulinastatin resulted in the inhibition of MMP-9 activity and plasmin activity in HCT 116 cells, as demonstrated by gelatin zymography and ELISA analysis. Moreover, the in vivo BALB/C nu/nu mice model demonstrated that ulinastatin significantly decreased the likelihood of recurrence following the surgical removal of hepatic metastases originating from colon cancer [182].

Marshal et al. described the creation of a powerful and extremely specific allosteric MMP-9 inhibitor, known as the humanized monoclonal antibody GS-5745. This inhibitor can be utilized to assess the effectiveness of MMP-9 inhibition as a treatment option for cancer patients. Their study demonstrates that specifically inhibiting MMP-9 did not cause musculoskeletal syndrome, which is a common side effect of broad-spectrum MMP inhibitors, in a rat model and, additionally, they discovered that inhibiting MMP-9 resulted in a decrease in tumor development and the occurrence of metastases in a surgical orthotopic xenograft model of CRC. Furthermore, inhibiting MMP-9 from either the tumor or the stroma was enough to prevent the growth of the primary tumor [183].

*KiSS-1* is a gene that encodes a protein called kisspeptin, which plays a crucial role in various physiological processes, particularly in the regulation of puberty and reproduction, and was originally identified as a metastasis suppressor gene [184]. Lentivirus infection was used to achieve stable transfection of *KiSS-1*-specific siRNA and *KiSS-1* expression vector in the human CRC cell line HCT 116. Overexpressing *KiSS-1* resulted in a substantial reduction in the proliferation and invasiveness of HCT 116 cells, while simultaneously enhancing apoptosis. The Western blotting analysis revealed that the expression of MMP-9 and the phosphorylation of Akt were markedly inhibited due to the overexpression of *KiSS-1* [185].

TIMP-2 plays a role in regulating the extracellular matrix by inhibiting MMPs, particularly MMP-2 and MMP-9, and by doing so, it helps maintain the structural integrity of tissues and prevents excessive degradation of the extracellular matrix [186]. Wang and colleagues assessed the expression of TIMP-2 and MMP-9 in a CRC tissue microarray using immunohistochemistry. Their findings indicate that decreased levels of TIMP-2 or increased levels of MMP-9 in cancerous tissues are associated with a reduced overall survival rate. The expression of TIMP-2 or MMP-9 was found to be an independent predictive marker for CRC. Moreover, the combined expression of TIMP-2 and MMP-9 played a synergistic function in accurately predicting the prognosis of patients with CRC. TIMP-2 has shown the ability to hinder the invasion and migration of HCT 116 cells in both laboratory and living organism settings by controlling the activity of MMP-9 [187].

N-Methylsansalvamide (MSSV) is a cyclic pentadepsipeptide that contains a sequence of five amino acids, where one of the amide bonds is replaced with an ester bond. This cyclic structure often contributes to its stability and biological activity and has been found to exhibit potent anti-cancer activity [188]. The suppression of cell growth in HCT116 cells was observed due to the induction of G0/G1 cell cycle arrest. An interesting study found that MSSV caused a decrease in MMP-9 levels by reducing the binding activity of AP-1, Sp-1, and NF-κB motifs. This decrease in MMP-9 levels inhibited the migration and invasion of HCT 116 cells. The introduction of MSSV via oral administration effectively suppressed the development of tumors in HCT 116 xenograft mice [188].

Protein- and peptide-based gelatinase inhibitors hold promise in CRC treatment, but they also come with several disadvantages and challenges. Proteins and peptides are susceptible to degradation by proteases in the gastrointestinal tract and bloodstream, which can reduce their effectiveness [189]. Maintaining the stability of these inhibitors during storage and upon administration can be difficult. Proteins and peptides are typically not well absorbed when taken orally, necessitating alternative routes of administration such as intravenous or subcutaneous injections [190]. They often have short half-lives in the bloodstream, leading to the need for frequent dosing, and proteins and peptides can induce immune responses, which can lead to adverse effects and reduced therapeutic efficacy [191].

Ensuring that gelatinase inhibitors specifically target cancer cells without affecting normal cells is challenging. Off-target effects can lead to toxicity and other side effects and effectively delivering these inhibitors to the tumor site while avoiding healthy tissues is a significant challenge. This often requires sophisticated delivery systems, such as nanoparticles or conjugation with targeting ligands, and achieving sufficient penetration into the dense extracellular matrix of tumors can be difficult [192,193]. The production of protein and peptide drugs can be complex and costly, involving sophisticated techniques like recombinant DNA technology and peptide synthesis, and scaling up the production while maintaining consistency and purity can be challenging [194]. The regulatory approval process for biologics is often more stringent and time-consuming compared to small-molecule drugs, requiring extensive testing for safety, efficacy, and quality [195].

Researchers are actively exploring various strategies to overcome these challenges, such as modifying peptides and proteins to enhance their stability and resistance to proteolytic degradation, developing nanoparticle-based delivery systems and conjugation with targeting ligands to improve bioavailability and target specificity, using gelatinase inhibitors in combination with other therapeutic agents to enhance efficacy and prevent resistance, and designing inhibitors that are less likely to provoke an immune response. Despite these challenges, the potential benefits of protein- and peptide-based gelatinase inhibitors in treating CRC continue to drive research and development in this area.

## 8. Conclusions

In summary, the exploration of natural, chemical small molecules, microRNA, and protein- and peptide-based gelatinase inhibitors has unveiled promising therapeutic avenues for CRC. Natural compounds, with their diverse mechanisms and minimal side effects, offer a rich reservoir of potential inhibitors. Chemical small molecules, through their ability to specifically target gelatinases, provide precision in treatment strategies. The regulatory role of microRNAs in gene expression adds a layer of control, enhancing the efficacy of existing therapies. Protein- and peptide-based inhibitors, with their high specificity and potency, present a robust approach to directly targeting gelatinase activity. Collectively, these gelatinase inhibitors show substantial potential in inhibiting CRC progression, metastasis, and enhancing patient survival rates. Future research should focus on optimizing the delivery methods, understanding the long-term effects, and combining these inhibitors with current treatment modalities to maximize therapeutic outcomes. By integrating these diverse strategies, a more effective and comprehensive approach to combating CRC can be achieved.

## Figures and Tables

**Figure 1 biomolecules-15-00035-f001:**
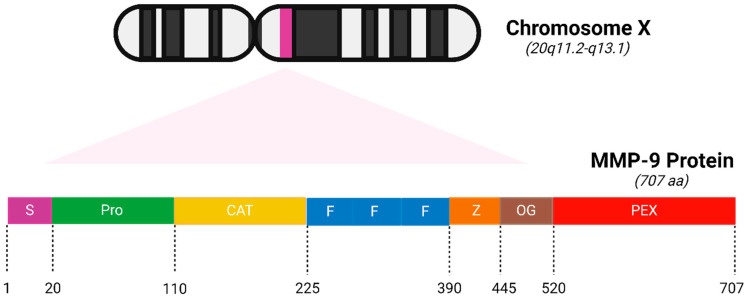
The multidomain structure of MMP-9. The following domains are shown: the signal peptide (purple), the propeptide (green), the catalytic domain (yellow), the three fibronectin repeats (blue), the metal binding site (orange), the OG domain (brown), and the PEX domain (red).

**Figure 2 biomolecules-15-00035-f002:**
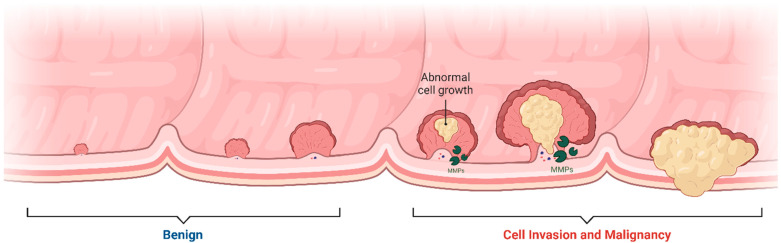
The image illustrates the transition from benign growths to malignant cell invasion in colorectal cancer, highlighting the abnormal cell growth and the presence of matrix metalloproteinases (MMPs), particularly MMP-2 and MMP-9, that facilitate cancer cell invasion. (Illustration created with BioRender.com).

**Figure 3 biomolecules-15-00035-f003:**
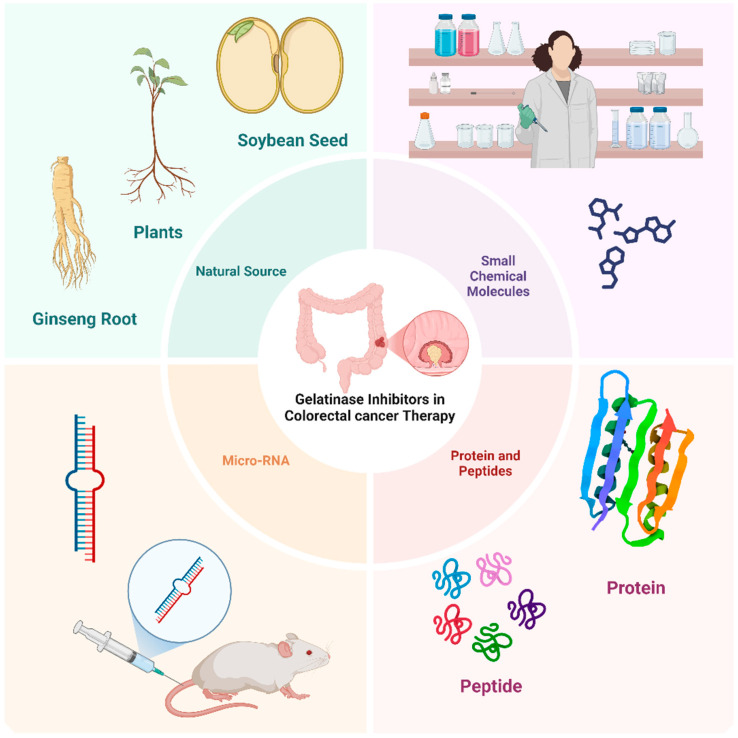
The image illustrates the comprehensive approaches for gelatinase inhibition in colorectal cancer therapy, encompassing natural sources, small chemical molecules, proteins, peptides, and microRNA-based strategies. (Illustration created with BioRender.com).

**Figure 4 biomolecules-15-00035-f004:**
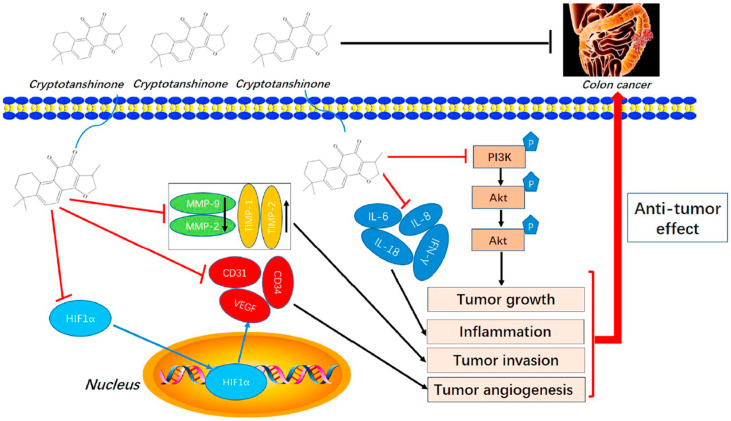
Schematic representation of the underlying process by which CPT inhibits CT26 colon cancer. CPT suppresses the production of MMP-2 and MMP-9 and enhances the production of TIMP-1 and TIMP-2, resulting in the prevention of tumor invasion by CPT. The CPT suppresses the expression of inflammatory factors, hence reducing inflammation and inhibiting tumor inflammation. In addition, the PI3K/Akt/mTOR pathway is crucial for tumor growth. CPT effectively hinders the phosphorylation of PI3K, Akt, and mTOR, hence suppressing tumor growth. These combined effects contribute to the anti-tumor impact of CPT. Adapted with permission from [63], Elsevier, 2018.

**Figure 5 biomolecules-15-00035-f005:**
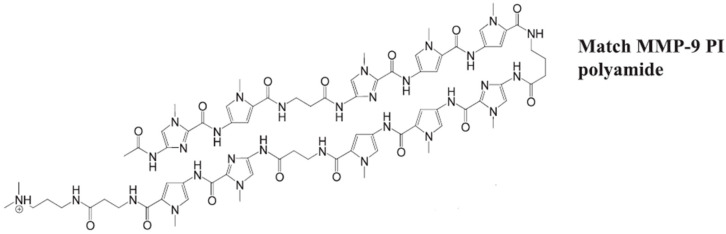
Structure of the human MMP-9-specific pyrrole-imidazole (PI) polyamide. Adapted with permission from [115], Wiley, 2010.

**Figure 6 biomolecules-15-00035-f006:**
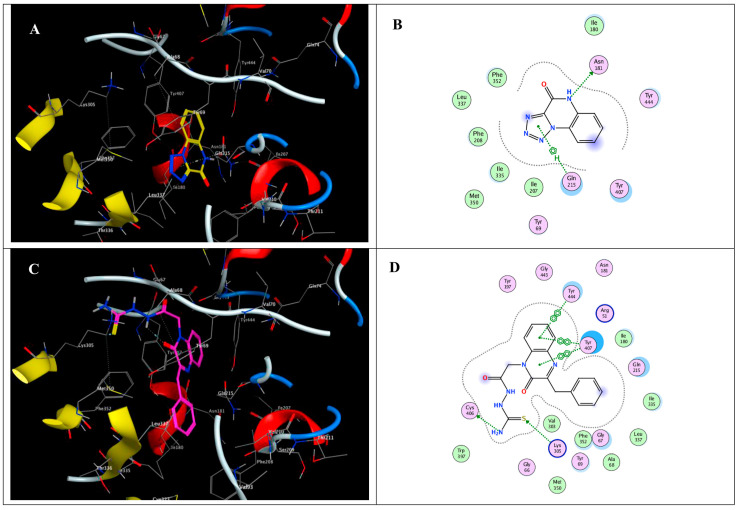
(**A**) 3D binding mode of 8 (yellow sticks), (**B**) 2D binding mode of 8, (**C**) 3D binding mode of 16 (magenta sticks), (**D**) 2D binding mode of 16 in the catalytic domain of MMP-9 (PDB ID: 1GKC). Adapted with permission from [126], Elsevier, 2021.

**Figure 7 biomolecules-15-00035-f007:**
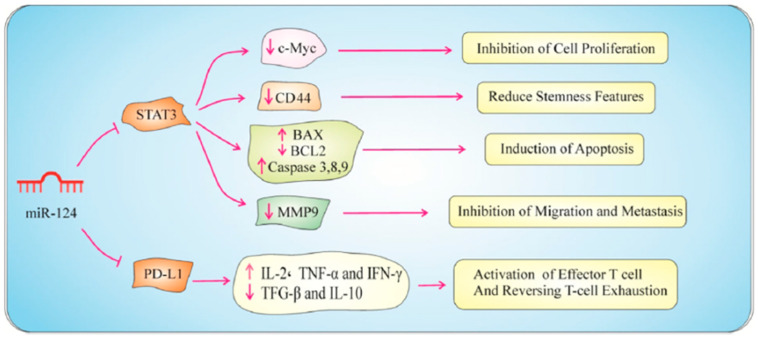
miR-124 specifically targets PD-L1 and STAT3. The STAT3 and PD-L1 pathways have a significant impact on the progression of colorectal cancer and the immunological response of T cells. Thus, by modifying downstream effectors like MMP-9, it is possible to limit CRC carcinogenesis through the regulation of PD-L1 and STAT3 expression and activity. Adapted with permission from [149], Wiley, 2010.

**Table 1 biomolecules-15-00035-t001:** List of natural-based compounds explored as gelatinases inhibitor in colorectal cancer.

Name	Type and Source	Structure	Cell Type	Function	Refs.
Myricetin	Plants	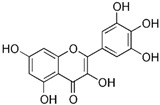	HT-29, COLO 205	Inhibitory effects on MMP-2 enzyme activity, Blocked TPA-stimulated invasion	[52]
EGCG	Phenol/green tea	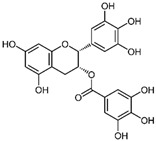	SW837, SW620	Significant reduction in the cellular levels of MMP-9 mRNAs, Suppressed the proliferation and migration	[53,54]
Curcumin	Turmeric	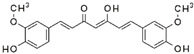	COLO 205, HCT 116, HT- 29, SW 620	Inhibiting MMP-2 and MMP-9 enzymes and blocking the invasion, Blocked MMP-9 activation and gene production by inhibiting the NF-κB, Reduced tumor volume in nude mice	[55,56]
Saponin	Soybean	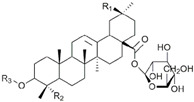	CT-26	Reduced mRNA expression and the secretion of MMP-2 and MMP-9, Moderate reduction in the occurrence of metastatic lung tumors in the mice	[57]
Norcantharidin	Mylabris phalerata Pall.	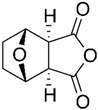	CT-26	Reduced MMP-9 mRNA and protein levels and its gelatinolytic activity	[58]
Anthocyanins	Plants	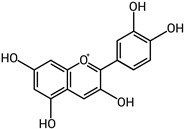	HCT 116	Anti-invasive properties on cancer cells, Suppression of MMP-2 and MMP-9 protein expression and activity	[59]
Aloe emodin	Aloe vera	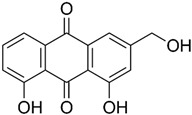	WiDr	Inhibited mRNA expression and gelatinolytic activity of MMP-2 and MMP-9, Blocking the PMA induced migration and invasion	[60]
Brefeldin A	Eupenicillium brefeldianum	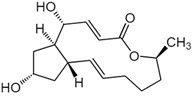	COLO 205	Significantly decreased the viability of cells by promoting apoptosis, Diminished the activity of MMP-9	[61]
Andrographolide	Andrographis paniculata	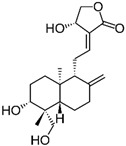	SW620	Inhibit cell proliferation, Enhance cytotoxicity, Induce apoptosis, Suppression of MMP-9 signaling pathways	[62]
Cryptotanshinone	Salvia miltiorrhiza	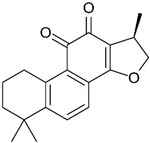	CT-26	Effectively inhibited cell invasion in vitro, Reduced the protein levels of MMP-2 and MMP-9, Increased TIMP-1 and TIMP-2	[63]
Rosmarinic Acid	Rosmarinus officinalis	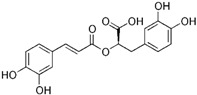	CT-26, HCT 116	Cell cycle arrest and apoptosis. Hindered the invasion and migration of cells, Decreased the levels of MMP-2 and MMP-9	[64]
Brassinin	Cruciferous vegetables	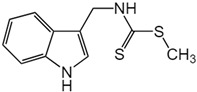	SW-480	Substantial reduction in MMP-9 activity, Induced apoptosis	[65]
Resveratrol	Plants	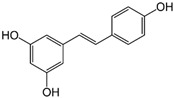	HCT 116	The anti-proliferative and anti-cancer properties related to MMP-9 reduction	[66]
Fucoxanthin	Brown seaweeds	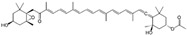	SW620	Significantly inhibited the proliferation of cells, Downregulation of MMP-9 mRNA and protein expression	[67]
Silibinin	Silybum marianum	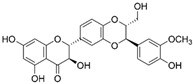	HT-29	Reduced the viability of cell, Significantly inhibited MMP-2 and MMP-9 mRNA and protein expression	[68]
Chrysin	Plants	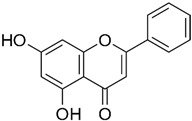	SW620	At dosages of 125 and 150 mg/kg, significantly reduced the MMP-9 levels2, Anti-invasion activity versus	[69]
Sauchinone	Saururus chinensis	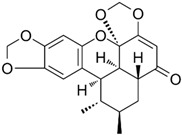	SW840, HCT 116	Decreased the expression of MMP2 and MMP9, Enhanced the cytotoxicity ofcells co-cultured with CD8+ T cells	[70]
Tannic Acid	Plants	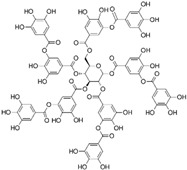	SW48	Suppressed the survival, colony formation, and migration, significantly reduced the levels of MMP-9 expression	[71]
Triptolide	Tripterygium wilfordii	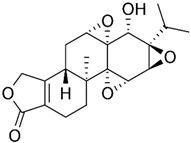	HT-29	Significantly decreased the proliferation and invasion of cells, Enhanced apoptosis, Reduced levels of MMP-2 and MMP-9	[72]
Punicalagin	Punica granatum	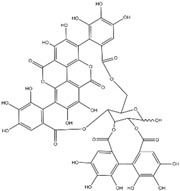	HCT 116, HT-29, LoVo	Cytotoxic effects on colon cancer cells, Suppression of MMP-2 and MMP-9 expression	[73]
Hawthorn Proanthocyanidin	Hawthorn	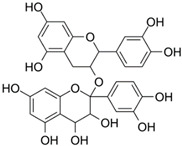	HCT 116	The levels of N-cadherin and MMP-9 were significantly reduced and led to inhibitory effect on cells migration	[74]

**Table 2 biomolecules-15-00035-t002:** List of synthetic small molecules explored as gelatinases inhibitor in colorectal cancer.

Name	Structure	Cell Type	Function	Ref.
MMI270	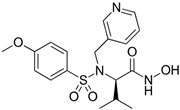	KM12SM	Significantly inhibited lung metastasis, Higher survival rates, Suppressed MMP-9 activities, Reduced the relative MMP-2 activity	[102]
Gefitinib	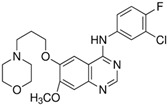	HT-29	Inhibited the secretion and mRNA expression of MMP-9 and MMP-2, Reduced the cells’ ability to adhere to laminin and type IV collagen	[103]
Etodolac	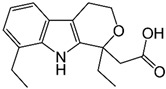	Colon 26	Reduced expression of MMP-9 mRNA, Decreased metastatic nodules on the liver	[104]
AMD3100	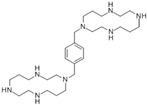	SW480	Cell viability blocked, Hindered the invasion ability, Significantly decreased the expression of MMP-9 but not MMP-2	[105]
17β-estradiol	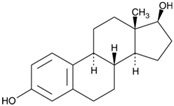	LoVo	Decreased MMP-2 and MMP-9 expression, Reduced cell mobility by suppressing activation of JNK1/2 and p38α MAPK signaling pathway	[106]
Volatile Anesthetics	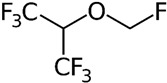	MC-38	Reduced MMP-9 release by IL-8-stimulated human neutrophils in vitro, Reduced mouse colon carcinoma cell migration across simulated extracellular matrix in vitro	[107]
Celecoxib	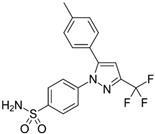	-	Decrease in gelatinases levels, TIMP-2 levels were significantly higher in the celecoxib-treated group	[108]
GL-V9	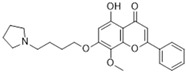	HCT116, SW480, SW620, LS174T	Reduces cell viability, migration, and invasion in a concentration-dependent fashion, Significantly decreased both the protein expression levels and activities of MMP-2 and MMP-9	[109]
β-Sitosterol	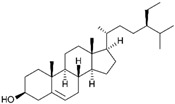	CT-26/luc cells	Effectively inhibited metastases, Reduced MMP-9 expression	[95]
SB202190	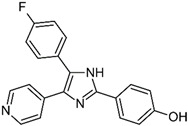	HT-29, SW480, SW620	Reduced levels of MMP-2 and MMP-9 and related invasiveness in cells	[110]

**Table 3 biomolecules-15-00035-t003:** microRNAs which indirectly impact gelatinases expression and activity in colorectal cancer cells.

Name	Target Gene	Cell Type	Function	Refs.
miRNA-34a	*Fra-1*	HCT 116	Significantly suppressed cell migration and invasion, The levels of MMP-9 were reduced	[136]
miRNA-22	*TIAM1/NLRP3*	HCT 116	Significantly reduced the viability and migration and invasion capabilities. Decrease in the levels of the pro-invasive MMP-2 and MMP-9 expression	[137,138]
miRNA-195	*CARMA3*	SW480, HT-29	Modulated MMP-9 via targeting CARMA3	[139]
miRNA-149	*FOXM1*	HCT 116, LoVo, SW480	mRNA expression levels of MMP-2 and MMP-9 were downregulated	[140]
miRNA-302a	*MAPK, PI3K/Akt*	SW620, SW480, HT-29, HCT 116	Inhibited the proliferation and invasion of cells, The expression and secretion of MMP-9 and MMP-2 were notably reduced	[141]
miRNA-206	*NOTCH3*	SW480, SW620	Prevented cancer cell proliferation and migration, Arrested the cell cycle, Triggered apoptosis, Inhibition of MMP-9 expression	[142]
miRNA 9	*TM4SF1*	SW480	Impaired transwell migration and invasion, Led to a decrease in MMP-2 and MMP-9 levels	[143]
miRNA-497	*Fra-1*	HCT 116, SW480	The levels of MMP-2 and MMP-9 proteins and invasion were significantly reduced	[144]
miRNA-7	*FAK*	HCT-8, Caco-2	Inhibited proliferation and migration of cells, impacted the expression of MMP-2 and MMP-9	[145]
miRNA-875-5p	*EGFR*	HCT 116, SW480	Reduced cell proliferation, Induced apoptosis, Impaired cellular migration and invasion by inhibiting MMP-9	[146]
miRNA-202-5p	*SMARCC1*	HCT 116, SW480	Suppressed cell growth and metastasis, Inhibited MMP-9	[147]
miRNA-128	*RPN2*	HT-29	Significantly suppressed cell proliferation, migration, and invasion, Led to lower levels of MMP-2 and MMP-9, while increasing levels of TIMP-2	[148]
miRNA-124-3p	*PD-L1*	HT-29, SW480	Proliferation of cells was reduced, Cell cycle was halted at the G1 phase, Reduced MMP-9 expression leading to the inhibition of cell motility and invasion	[149]

## Data Availability

Not applicable.

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
