# Peer review of "Targeting Invasion: The Role of MMP-2 and MMP-9 Inhibition in Colorectal Cancer Therapy"

_biomolecules, 2024, doi:10.3390/biom15010035_

Round 1
Reviewer 1 Report
Comments and Suggestions for Authors
The review entitled " Targeting Invasion: The Role of MMP-2 and MMP-9 Inhibition in Colorectal Cancer Therapy " addresses a highly topical scientific issue, considering the role of gelatinases in cancer and, specifically, in colorectal cancer (CRC).
The manuscript provides an in-depth discussion of both natural and synthetic inhibitors, describes the significance of miRNAs, and extends to Protein- and Peptide-based Inhibitors, offering a detailed overview of potential future strategies. These approaches could serve as alternatives or complements to current antitumor therapies.
The structure of the text is clear and well-organized, with figures and tables effectively summarizing the topics covered. The bibliography is up-to-date and consistent with the content of the manuscript.
There are minor typographical errors and misspellings that should be corrected in the final draft.
In my opinion, the work aligns perfectly with the scope of the special issue “Matrix Metalloproteinases: Mechanisms, Functions and Therapeutic Potential” and can be published in its current form following minor corrections.
Minor revision.
Lane 173: kinds of cancer
Lane 187: for example
Lane 222: involves
Lane 230: collegues
Lane 239: 1,2-dimethylhydrazine
Lane 253: cell migration
Lanes 593-595: MMP-9 and MMP-2 expression?
Lane 614: epidemiological
Lane 781: the upregualtion
Lane 783: SW620.
LANES 789-780: It has been shown that
Lane 801: In a study
Lane: 807: where their findings
Lane 969: Their findings
Lane 974: has shown
Lanes 982-983: it is not clear: This decrease in MMP-9 levels resulted in the migration and invasion REDUCTION (?)
Lane 983: “via oral administration” is better than “through the mouth”
Author Response
Dear Editor,
We sincerely thank you and the reviewers for the insightful comments and constructive feedback on our manuscript entitled "Targeting Invasion: The Role of MMP-2 and MMP-9 Inhibition in Colorectal Cancer Therapy." We are pleased to submit the revised version of our manuscript, incorporating the suggestions provided.
Below, we address each comment raised by the reviewer, detailing the changes made in the manuscript.
Reviewer 1 Comments and Our Responses
- Lane 173: "kinds of cancer"
Response: Corrected to "kinds of cancer" in the revised manuscript. - Lane 187: "for example"
Response: Incorporated "for example" to improve clarity as suggested. - Lane 222: "involves"
Response: Corrected the text to use "involves" appropriately in this context. - Lane 230: "colleagues"
Response: Corrected the typographical error to "colleagues." - Lane 239: "1,2-dimethylhydrazine"
Response: Corrected the chemical name to "1,2-dimethylhydrazine." - Lane 253: "cell migration"
Response: Corrected the phrase to "cell migration." - Lanes 593–595: "MMP-9 and MMP-2 expression?"
Response: Clarified the sentence to “In similar way, aloe emodin, a natural anthraquinone, inhibited the nuclear translocation and DNA binding of NF-κB and downregulated mRNA and consequently de-creased expression and promoter/gelatinolytic activity of MMP-2 and MMP-9 which led to blocking the phorbol-12-myristyl-13-acetate (PMA) induced migration and invasion of WiDr colon adenocarcinoma cells” - Lane 614: "epidemiological"
Response: Corrected the spelling to "epidemiological." - Lane 781: "the upregulation"
Response: Corrected the phrasing to "the upregulation." - Lane 783: "SW620"
Response: Corrected the formatting to accurately refer to "SW620." - Lanes 789–780: "It has been shown that"
Response: Corrected the phrasing to "It has been shown that." - Lane 801: "In a study"
Response: Corrected the phrasing to "In a study." - Lane 807: "where their findings"
Response: Adjusted the sentence for clarity: "where their findings" - Lane 969: "Their findings"
Response: Corrected the phrasing to "Their findings." - Lane 974: "has shown"
Response: Adjusted to "has shown" to ensure grammatical accuracy. - Lanes 982–983: "This decrease in MMP-9 levels resulted in the migration and invasion REDUCTION (?)"
Response: Clarified the sentence to read: " This decrease in MMP-9 levels inhibited the migration and invasion of HCT 116 cells." - Lane 983: "via oral administration" is better than "through the mouth."
Response: Revised the text to use "via oral administration" for improved scientific clarity.
We have also carefully reviewed the manuscript for additional typographical errors, ensuring consistency and clarity throughout.
Thank you for your consideration of our revised manuscript. We are confident that the changes have enhanced the quality of the work and addressed the reviewer’s concerns thoroughly. We look forward to your feedback and hope for a favorable decision on the publication of our manuscript.
Sincerely,
Alireza Shoari, PhD
Reviewer 2 Report
Comments and Suggestions for Authors
This Review highlights the critical roles of gelatinases (MMP-2 and MMP-9) in colorectal cancer (CRC) progression, emphasizing their potential as therapeutic targets. It outlines innovative inhibition strategies, including small molecule inhibitors, natural compounds, and gene silencing, while addressing challenges like nonspecificity and resistance. The integration of these inhibitors with conventional and emerging therapies offers a promising path to enhance efficacy and overcome resistance. Overall, the Review underscores the clinical relevance of targeting gelatinases and inspires further research to improve CRC outcomes.
This Review highlights the therapeutic potential of targeting gelatinases to improve colorectal cancer outcomes, addressing critical challenges in treatment efficacy and resistance. It can be published after minor revision.
Abstract:
Please, highlight the Aim of the Review early in the text.
Please, write a conclusion with clear explanation on novelty of the review and its potential benefits for CRC prevention and therapy.
Main text:
To ease perception, please, shorten textual description of all chapters describing various types of MMP inhibitors. Remove all irrelevant fragments. It also would be good to improve cohesiveness and logic flow.
It also would be good to prepare a figure unifying and illustrating various types of MMP inhibitors.
These minor concerns in no way diminish the value of this review.
Author Response
Dear Editor,
We sincerely thank you and the reviewers for the insightful comments and constructive feedback on our manuscript entitled "Targeting Invasion: The Role of MMP-2 and MMP-9 Inhibition in Colorectal Cancer Therapy." We are pleased to submit the revised version of our manuscript, incorporating the suggestions provided.
Below, we address each comment raised by the reviewer, detailing the changes made in the manuscript.
Reviewer 2 Comments and Our Responses
Abstract
- Highlight the Aim of the Review early in the text.
Response: The abstract has been revised to explicitly state the aim of the review in the opening sentences to provide a clear focus on the objectives of the work. - Write a conclusion with a clear explanation of the novelty of the review and its potential benefits for CRC prevention and therapy.
Response: A concluding section has been added to the abstract, emphasizing the novelty of our review and its potential impact on CRC prevention and therapeutic strategies.
Main Text
- Shorten textual descriptions of all chapters describing various types of MMP inhibitors. Remove all irrelevant fragments.
Response: The sections discussing different MMP inhibitors have been reviewed and condensed. Redundant and irrelevant content has been removed to improve clarity and brevity. - Improve cohesiveness and logical flow.
Response: We have reorganized the text to enhance the logical flow between sections, ensuring a smoother transition and more cohesive narrative throughout the manuscript. - Prepare a figure unifying and illustrating various types of MMP inhibitors.
Response: A new figure has been designed and included in the manuscript, summarizing and categorizing the different types of MMP inhibitors. This visual aid provides a comprehensive and unified representation of the content for easier understanding.
We have also carefully reviewed the manuscript for additional typographical errors, ensuring consistency and clarity throughout.
Thank you for your consideration of our revised manuscript. We are confident that the changes have enhanced the quality of the work and addressed the reviewer’s concerns thoroughly. We look forward to your feedback and hope for a favorable decision on the publication of our manuscript.
Sincerely,
Alireza Shoari, PhD